# Identification of Kinetic Abnormalities in Male Patients after Anterior Cruciate Ligament Deficiency Combined with Meniscal Injury: A Musculoskeletal Model Study of Lower Limbs during Jogging

**DOI:** 10.3390/bioengineering9110716

**Published:** 2022-11-19

**Authors:** Shuang Ren, Xiaode Liu, Haoran Li, Yufei Guo, Yuhan Zhang, Zixuan Liang, Si Zhang, Hongshi Huang, Xuhui Huang, Zhe Ma, Qiguo Rong, Yingfang Ao

**Affiliations:** 1Beijing Key Laboratory of Sports Injuries, Department of Sports Medicine, Institute of Sports Medicine of Peking University, Peking University Third Hospital, Beijing 100191, China; 2Department of Mechanics and Engineering Science, College of Engineering, Peking University, Beijing 100871, China; 3Intelligent Science & Technology Academy of CASIC, Beijing 100043, China

**Keywords:** anterior cruciate ligament deficiency, meniscal injury, jogging, knee kinetics, inverse dynamics, musculoskeletal model

## Abstract

There is little known about kinetic changes in anterior cruciate ligament deficiency (ACLD) combined with meniscal tears during jogging. Therefore, 29 male patients with injured ACLs and 15 healthy male volunteers were recruited for this study to investigate kinetic abnormalities in male patients after ACL deficiency combined with a meniscal injury during jogging. Based on experimental data measured by an optical tracking system, a subject-specific musculoskeletal model was employed to estimate the tibiofemoral joint kinetics during jogging. Between-limb and interpatient differences were compared by the analysis of variance. The results showed that decreased knee joint forces and moments of both legs in ACLD patients were detected during the stance phase compared to the control group. Meanwhile, compared with ACLD knees, significantly fewer contact forces and flexion moments in ACLD combined with lateral and medial meniscal injury groups were found at the mid-stance, and ACLD with medial meniscal injury group showed a lower axial moment in the loading response (*p* < 0.05). In conclusion, ACLD knees exhibit reduced tibiofemoral joint forces and moments during jogging when compared with control knees. A combination of meniscus injuries in the ACLD-affected side exhibited abnormal kinetic alterations at the loading response and mid-stance phase.

## 1. Introduction

Anterior cruciate ligament (ACL) and meniscus ruptures are common sports-related injuries and are often accompanied by a higher risk of osteoarthritis [1] even after ACL reconstruction [2,3]. The alterations in injured knee kinematics (such as joint angles and displacements), kinetics (such as joint moments, contact forces and ground reaction forces), and muscle activity are one of the reasons that lead to knee instability, thus causing secondary osteoarthritis. Numerous studies have investigated the characteristics of kinematics and kinetics in anterior cruciate ligament deficiency (ACLD) knees and found reductions in knee flexion moment and peak knee flexion angle, with patients adopting quadriceps avoidance [4] and stiffening strategy [5] gait patterns during level walking. For jogging, many studies focused on knee mechanics after ACL reconstruction. A study [6] reviewed the running biomechanics in patients with ACL reconstruction and pointed out that knee kinematics and kinetics were sensitive to sagittal plane motions. Asaeda et al. [7] investigated the relationship between knee muscle strength and knee biomechanics and found that both males and females were unable to restore their normal knee biomechanics from 3 months to 5 years. Moreover, some researchers [8,9,10] studied the knee mechanic alterations in ACLD patients during running and showed that kinematics changed both in the injured and uninjured knees. However, the effect of meniscus injuries on the ACLD knees during jogging is rarely reported.

On the other hand, accurate determination of kinetics involves difficulties and limitations in both in vitro cadavers and in vivo imaging studies. Musculoskeletal (MS) modeling could circumvent such shortcomings. Recently, many studies have performed multi-body dynamics modeling related to ACLD patients by using the musculoskeletal (MS) model and motion analysis system to estimate the joint and muscle forces. For instance, simulations were developed to predict ACL graft attachment locations during reconstruction [11], estimate knee contact mechanics and secondary kinematics with or without menisci [12], and evaluate the tibiofemoral joint forces of gait after ACL reconstruction [13]. Comparatively, there is little data on the evaluation of the kinetics of combined ACLD and meniscal injuries by using the MS model during jogging.

The aim of this study was to simulate the three-dimensional (3D) kinetics of ACLD knees with or without a medial or/and lateral meniscal injury during jogging by using the motion capture system and subject-specific multi-body dynamic model. The results could provide suggestions for clinical rehabilitation of muscle strength training and neuromuscular exercise. Our hypotheses were (1) the injured sides in ACLD knees would present significantly lower forces and moments during the stance phase, and the uninjured sides would be similar to healthy people; (2) ACLD combined with meniscal injury knees would show decreased moments during the stance phase compared with isolated ACLD knees.

## 2. Materials and Methods

### 2.1. Subjects

Twenty-nine young male patients with unilateral chronic ACLD knees (contralateral side intact) were recruited before undergoing ACL reconstruction. Ethical approval was obtained from the university’s ethics committee and written informed consent was attained from all subjects. Twenty-three patients were injured through a non-contact accident, and 6 patients were through a contact accident. Most of the injury accidents occurred during basketball and football. Their activity level was evaluated by the Tegner score, which is a reliable and valid instrument to evaluate the activity level of patients with ACL injuries in the Chinese population [14]. The activity level of all the patients was normal before knee injuries (score range 3.0–6.0, Table 1). Among these patients, 12 patients had isolated unilateral ACL injuries (ACLD group), 5 had combined ACL and lateral meniscal injuries (ACLDL group), 5 had combined ACL and medial meniscal injuries (ACLDM group), and 7 had combined ACL and medial/lateral meniscal injuries (ACLDML group). The ACL injuries were documented by MRI and clinical examination and confirmed by arthroscopic findings, ranging from 6 months to 4 years prior to testing. Exclusion criteria were that patients had no previous ACL and concomitant meniscus and ligament ruptures, as well as no history of musculoskeletal pathologies of hip or ankle joints. Eleven lateral menisci were posterior longitudinal tears, and one was an anterior horn tear. All the medial meniscus were posterior complex horn tears. These patients had failed non-operative treatment of the injury and were therefore recognized as ACLD non-coppers. Fifteen healthy volunteered for this study (Control group). All the participants were male in this study because biomechanical characteristics were different between genders [9]. The morphological data are shown in Table 1. No significant differences were found between each of the two groups in terms of age, height, weight, body mass index (BMI) and pace. The time since injury for all the participants was over 6 months.

### 2.2. Data Collection and Modeling Analysis

Between January 2014 and December 2016, the experimental data was collected by an 8-camera motion capture system at a sample rate of 100 Hz (Vicon MX; Oxford Metrics, Yarnton, Oxfordshire, UK), where the marker trajectory data were filtered at 12 Hz. The ground reaction forces were collected by two embedded force plates at a sampling rate of 1000 Hz (AMTI, Advanced Mechanical Technology Inc., Watertown, MA, USA). A series of 14 mm markers were attached to the anatomical lower limb locations based on the plug-in-gait model to track the segmental motion during jogging (Figure 1). The participants were asked to run along a 10-m path at a self-selected speed where the kinematic data were recorded by eight cameras. No participants complained about pain during jogging. For each subject, five successful running trials were recorded, and these results were imported into the multi-body dynamics software AnyBody Modeling System (version 6.0.5, AnyBody^TM^ Technology, Aalborg, Denmark) to estimate kinetics of the knee joint. The method of combining a motion capture system and musculoskeletal model in AnyBody Modeling System for gait analysis has been clinically validated in many studies, such as knee osteoarthritis [15,16], muscle activity [17,18] and kinematic characteristics of children’s gait [19].

The Twente Lower Extremity Model [20] implemented in the AnyBody Modeling System was employed for the analysis. This model was comprised of 12 body segments, and 6 joint degrees of freedom were considered for each leg, with a spherical joint with 3 degrees of freedom for the hip joint and a universal joint with 2 degrees of freedom for the ankle joint. The knee joint was modeled as a hinge joint with 1 degree of freedom due to the soft tissue artifact error [21]. Based on the morphological parameters, each model was scaled with a mass-fat scaling algorithm to perform the subject-specific jogging simulation. The min/max recruitment principle solver [22], which has better numerical convergence and physiological representation, was used to predict the muscle forces during the inverse dynamics analysis. The objective function minimizes the maximal muscle activity and can be formulated as follows:(1)$$\mathrm{Minimize}\;\max\left(\frac{f_{i}^{\left(M\right)}}{N_{i}}\right)$$

Subject to
(2)$$Cf=d,\;\;0\leq f_{i}^{\left(M\right)}\leq N_{i},\;\;\;\;i\in \left\{1,\ldots ,n^{\left(M\right)}\right\}$$
where $f_{i}^{\left(M\right)}$ and $N_{i}$ refer to the muscle forces and the strength of the muscles, respectively. f contains muscle forces $f^{\left(M\right)}$ and joint reactions $f^{\left(R\right)}$. C is the coefficient-matrix and the right-hand side. d contains applied loads and inertia forces.

Five different running trials were simulated based on the experimental data, and the average values were used to perform kinetic analysis by using MATLAB (version: 2016b, MathWorks, Natick, MA, USA).

### 2.3. Statistics

The 3D tibiofemoral joint kinetics during jogging were normalized to 0–100% gait cycle at 1% intervals. Analysis of covariance with jogging speed as a covariate was performed among participant characteristics and kinetics points among the five groups. A priori sample calculation was conducted before statistical analyses by using the effect size Cohen’s d. The forces and moments were analyzed by using a 1-way analysis of variance (ANOVA). A post hoc pairwise comparison using the Tukey multiple comparisons test was conducted when a significant difference was detected. For the interpatient kinetic differences of ACLD patients, two-way ANOVA was used to compare the tibiofemoral forces and moments among the ACLD-affected groups. The 2 within-patient factors were injury types (ACLD or ACLDL/ACLDM/ACLDML) and specific timings in the stance phase (10%, 20%, 30%, 40%, and 50% of gait cycle). The unpaired *t*-tests (independent sample) were performed between groups when the *F* ratio was at a significant difference level (*p* < 0.05).

## 3. Results

The jogging speed showed little effect on the kinetic parameters (*p* > 0.05). However, the injured and contralateral uninjured ACLDML knees showed large effect sizes in contact forces and moments compared to control knees (Table 2). Some gait parameters on the uninjured side showed large differences when there was a combined meniscal tear (ACLDL and ACLDM, d > 0.8 for ML, AP forces and axial moment). In particular, all the patients’ knees exhibited large differences in the minimum flexion moment measure (Table 2). No large effect sizes except flexion moment were found between the ACLD and control groups.

Decreased knee joint forces and moments of both legs in ACLD patients were detected during the stance phase when compared to the control group, such as mediolateral force (Figure 2A,G), anteroposterior force (Figure 2C,I), axial moment (Figure 2D,J) and flexion moment (Figure 2F,L). In particular, the ACLD knees with lateral and medial meniscal deficiency, both for injured and uninjured knees, showed significant differences from the control knees at mid-stance as for the kinetic parameters measure (Table 3). All the ACLD patients, with or without meniscal injury, ran with lower extension moments than the control groups (Figure 2F,L). Meanwhile, significant differences in minimum flexion moment were observed both in injured knees (ACLD: −0.82 ± 0.46 Nm/(Bw × Bh); ACLDML: −0.68 ± 0.37 Nm/(Bw × Bh); *p* < 0.05) and uninjured knees (ACLDL: −0.85 ± 0.30 Nm/(Bw × Bh); ACLDM: −0.73 ± 0.21 Nm/(Bw × Bh); *p* < 0.05) during the mid-stance (control: −1.21 ± 0.21 Nm/(Bw × Bh)) (Table 3). No significant differences in parameter measure of joint reaction forces were found among control, ACLD, ACLDL, and ACLDM, except for the comparison of anteroposterior forces between control and uninjured knee in ACLDL patients (Table 3).

The result of interpatient differences with a significant F ratio (group difference: *p* = 0.034) showed that significant differences were observed at the mid-stance phase (20% of the gait cycle) between ACLD and ACLDML groups. The ACLDML knees presented significantly less contact force as well as less flexion moment compared to the ACLD groups. It was also noted that ACLDM knees showed lower axial moment compared with ACLD knees in the loading response (10% of the gait cycle) (*p* < 0.05). Kinetics in ACLDL knees did not show any significant difference compared to the other patient groups.

## 4. Discussion

This study investigated the kinetic abnormalities in male patients after ACLD combined with meniscal injuries during jogging. The results showed that forces and moments of uninjured side in ACLD combined with meniscus injury patients were lower compared to the control legs and were similar to those of the injured side during the stance of jogging trials. Therefore, this result partly supports our first hypothesis. Differences among these patient groups showed that the varying effects on the knee joint kinetics were dependent on the type of meniscal injuries. ACLD patients combined with higher severity degrees of meniscus tears (e.g., ACLDML vs. ACLDL) exhibited more abnormal knee joint contact forces and moments. These results supported our second hypothesis, which also confirmed previous findings that described the abnormal knee biomechanics after ACL and meniscal injuries during walking [16,23] and ascending stairs [24].

Some studies have found abnormal changes in ACLD [8] and reconstruction [3,6,7] knees during running. However, they did not report the alteration of contact forces, which is closely related to the development of osteoarthritis. A few works of literature investigated the knee contact mechanics involving the function of menisci [13] and ACL reconstruction [11,12] during walking by using MS modeling, but none have mentioned the biomechanical characteristics while jogging. As far as we know, this is the first study to estimate the kinetics of ACLD knees with concomitant medial or/and lateral meniscal tears during jogging by using the multi-body dynamics model. Based on our results, all the ACLD knees, either with or without concomitant meniscus injuries, present little discrepancy compared to healthy volunteers during the swing phase, which was in line with a systematic review [6]. Thus, this study focused on the different kinetic alterations during the stance phase of the gait cycle among control limbs and injured and uninjured limbs in ACLD combined with meniscal injuries.

Existing literature on the knee joint kinetic analysis during running mainly concentrated on isolated ACLD injuries, and the results were inconsistent due to different methodologies [8]. Although it was difficult to make a direct comparison with current studies, a similar trend towards decreased knee joint kinetics was detected both in contact forces [25] and moments [26] after ACL reconstruction. Meanwhile, the flexion moment measure was similar to the result described by Pratt et al. [26] when the moment peak was normalized to body weight between 10% and 50% of the stance phase (Pratt approximately 3.1 Nm/(Bw) vs. this work 2.1 Nm/(Bw)). The difference may result from the running speed, in which the velocity of the control group is lower in this study, 2.33 m/s, compared to the author’s report of 4.24 m/s. The knee rotational moment was reported by some studies, which found a lower rotational moment in ACLD [8] and ACL reconstruction [27] knees compared to healthy limbs. The range of knee rotational moment for the control group compared favorably (0.07–0.13 Nm/(Bw × Bh) in [8] vs. 0.04–0.10 Nm/(Bw × Bh) in this study), and the peak value was similar to [27] when changing the unit $\mathrm{Nm}/\left(\mathrm{Bw}\cdot \mathrm{Bh}\right)$ to $\mathrm{Nm}/\left(\mathrm{Bm}\cdot m\right)$ (body mass and leg length). Comparable kinetic results to ours have also been reported by Sasimontonkul et al. [28]. The joint reaction force on the tibia was calculated by the inverse dynamic method, and the average axial and shear force peaks were 8.0 Nm/(Bw) and 1.2 Nm/(Bw), respectively. In this study, the proximo-distal and mediolateral force peaks were 8.6 Nm/(Bw) and 2.4 Nm/(Bw), respectively. Different modeling methods may account for this discrepancy. In the literature [28], the model was composed of four segmented models and 21 simplified muscles, and the sum of the cubed muscle stresses was used to calculate muscle forces. In this study, the lower extremity model consists of 12 body segments and 55 muscle-tendon units, and the muscle recruitment principle solver minimizes the maximal muscle activity. All these comparisons confirm the correctness of this study, and our data provide additional findings in kinetic alterations of ACLD combined with different types of meniscal injury during jogging.

### 4.1. Kinetics of Injured and Uninjured Knees

The uninjured side of the ACLD patients, either with or without meniscal injury, presented similar knee contact forces and moments compared to the injured side. However, knee contact forces and moments kinetic measures of both injured and uninjured legs for ACLD patients were lower when compared to the control group. This result was consistent with some studies developed by ACLD and ACL reconstruction subjects. They reported that the subjects continued to present decreased knee moments after ACL injury and reconstruction [25,27] during early stance [29] and weight acceptance [30], even 6 and 12 months after ACL reconstruction [7]. However, there is limited evidence about the tibiofemoral joint contact forces in individuals with ACLD combined with meniscal injuries during jogging.

The kinetic alterations are often associated with the compensatory mechanisms in ACLD knees. Papadonikolakis et al. [31] summarized some in vivo biomechanical investigations and showed that ACLD patients adapted to ACL injury by different strategies over a long period of time. Despite disagreements over adaptation mechanisms by which type of ACLD patients adopt, in order to keep the knee joint stable during jogging, ACLD subjects tend to reduce the knee moments mainly through three ways: (1) increasing the hamstring activity, (2) decreasing gastrocnemius activity and (3) avoiding quadriceps contraction. Based on previous studies, it is likely that an adaptation strategy occurs due to a long time of repetitive activities after ACL injuries. The participants in this study all experienced a long time of adaptations since ACL and meniscal injuries (>6 months). Some kinetic changes in both limbs by the neural system have developed during the daily motions in order to restore functional gait. In our opinion, long-term loss of ACL and meniscus contributed to the decreased kinetics in the knee joint. The abnormal biomechanical [1,26] associated with muscle activity changes [7,27] in ACLD combined with meniscal injury patients increased the risk of osteoarthritis as well as the incidence of chondral and meniscus pathology. Previous studies have shown that joint unloading was associated with the cascade of early degenerative changes at the knee joint [1]. The lower compressive forces in ACLDML knees maybe result in a higher incidence of cartilage degeneration. The less knee flexion moment in ACLD combined with meniscal injury groups could be explained as a protective adaptation strategy to avoid excessive anterior tibial displacement. As the average maximum anterior displacement and external rotation in ACLD knees were significantly less than that of the contralateral healthy knee [32], lower flexion moment in ACLD combined with the meniscal injury group during the stance phase might also be an adaptation to the muscle firing patterns. It tends to reduce the flexion moment and increase the net quadriceps moment and the activity of the hamstrings to achieve a more normal tibiofemoral position during jogging. The lower flexion moment detected in ACLD, ACLDM, and ACLDML groups (Figure 2F, L) confirms this muscle co-contraction strategy. Subjects after ACL and meniscus injuries consistently stabilize their knee with a stiffening strategy [5,23] involving less knee flexion motion and higher muscle contraction.

From a clinical point of view, neuromuscular training for the ACLD, ACLDM, ACLDL, and ACLDML patients during jogging needs to be provided to restore normal knee biomechanics and thus reduce the risk for secondary cartilage injury and osteoarthritis. The ACLDML patients presented more alterations with the control group during jogging than the ACLDM and ACLDL patients. The ACLDML patients presented lower moments and contact forces in the sagittal, coronal and axial planes for both injured and uninjured legs. Therefore, ACLDML patients should enhance knee extension, abduction, and rotation muscle strength for both the injured and uninjured legs. The uninjured knees of the ACLDM and ACLDL patients showed lower rotation moments during jogging than the control group. The ACLDM and ACLDL patients should pay attention to the rotation muscle strength training of the uninjured legs. The injured knees of the ACLD patients presented lower rotation moments than control knees, which implies that the ACLD patients should enhance rotation muscle strength for the injured legs. Muscle strength and neuromuscular training should be conducted after ACL rupture, and normal joint kinetics should be one standard for returning to jogging.

### 4.2. Interpatient Differences of ACLD Knees with or without Meniscal Injury

Interpatient differences in ACLD knees, with or without meniscal injury, have been reported by many studies [16,23,24,33,34]. Similar to our group category, Zhang et al. performed two studies investigating kinematic characteristics during level walking [33] and ascending stairs [24]. Various effects of meniscus injuries on knee joint stability showed that a combined ACL and meniscal injuries could alter the biomechanical response in different ways, depending on the type of meniscal tears. Ren et al. [23] demonstrated a combination of “stiffening gait” and “pivot shift gait” patterns in ACLD knees with medial meniscus posterior horn tear, and this abnormal gait was confirmed by Liu et al. [16] in ACLD combined with medial and lateral meniscus injury patients. However, these studies mainly focused on the measurement of kinematics and kinetics during gait [16,23,33] and ascending stairs [24,34] rather than running.

Previous studies have shown that meniscal behavior showed characteristics of non-linearity, anisotropy, and non-homogeneity [35], resulting in complex kinetic alterations after meniscus injuries. In this study, we found that medial meniscus status was an important factor in the kinetic alterations of ACLD knees, as the ACLDML group presented significantly decreased kinetics at the mid-stance (20% of the gait cycle), and the ACLDM group showed fewer axial moments during the loading response (10% of the gait cycle) when comparing to ACLD group. Different functions of the meniscus may provide an explanation of the result. The medial meniscus was closely related to the sagittal tibial translation, while the lateral was more susceptible to transverse and frontal movements, playing an important role in postural stability [36]. Meanwhile, it was reported that most meniscal tears were peripheral posterior horn, and the peripheral meniscal tear in the medial meniscus contributed to the intense contraction of the hamstring muscles [37]. Medial meniscal injuries thus may result in more movement disorders when performing the simulation. However, further studies with more fundamental activities in ACLD combined with meniscal injury patients, such as pivoting [8] and cutting [10], are needed to confirm the effect of different medial and lateral meniscal injuries on the knee joint biomechanics.

Some limitations of this study should be noticed. First, the minimum number of subjects among these groups is small (*n* = 5), and it is not convincing to represent a whole population of individuals with different meniscus injury patterns during jogging. Second, individual movement alterations were evaluated only by kinetics, and the knee joint was modeled as a hinge joint due to soft tissue artifact error [21]. Muscle activity and small motions of kinematics, such as rotations in the transverse plane and translations in the coronal plane, were not considered in this study. Continued studies of a large number of subjects with a more comprehensive parameter analysis, for instance, muscle forces and ground reaction forces, will be necessary to confirm further the adaptation strategy in the patient who combined ACL and meniscus injuries before ACL reconstruction.

## 5. Conclusions

Using a motion capture system and subject-specific musculoskeletal models of lower limbs, this study first estimated tibiofemoral joint kinetic abnormalities in male patients of ACLD knees with or without a concomitant meniscal injury during jogging. The results demonstrated that a combined ACL and meniscal injury could alter kinetics of lower limbs during the stance phase depending on the presence and type of meniscal tears. Compared to the isolated ACL injuries, a combination of meniscus injuries in the ACLD-affected side would exhibit more abnormal alterations at the loading response phase (for the ACLDM group) and mid-stance phase (for the ACLDML group) during jogging. These abnormal alterations will result in the instability of the knee joint and increase the risk of degenerative changes and the occurrence of knee osteoarthritis. Our musculoskeletal model analysis of lower limbs also presented the necessity to implement musculoskeletal modeling of increasing complexity to reveal biomechanical characteristics and ultimately provide help for personalized rehabilitation schemes. Further studies should focus on the relationship between long-term biomechanical changes and the development of protective mechanisms after ACL and meniscal injuries, as well as subsequent specific treatment and rehabilitation.

## Figures and Tables

**Figure 1 bioengineering-09-00716-f001:**
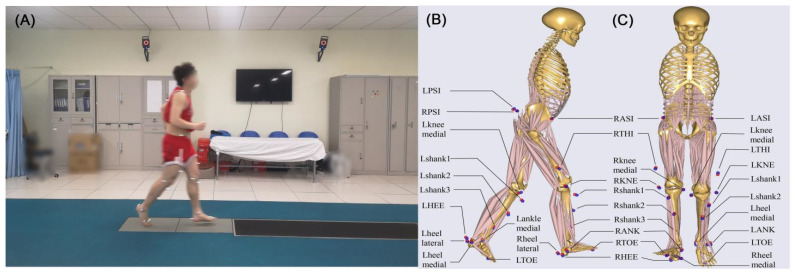
(**A**) Experimental data collection diagram; (**B**) The side view and (**C**) the front view of locations and names of the designated (red)/experimental (blue) markers on the musculoskeletal model when performing inverse dynamic analysis.

**Figure 2 bioengineering-09-00716-f002:**
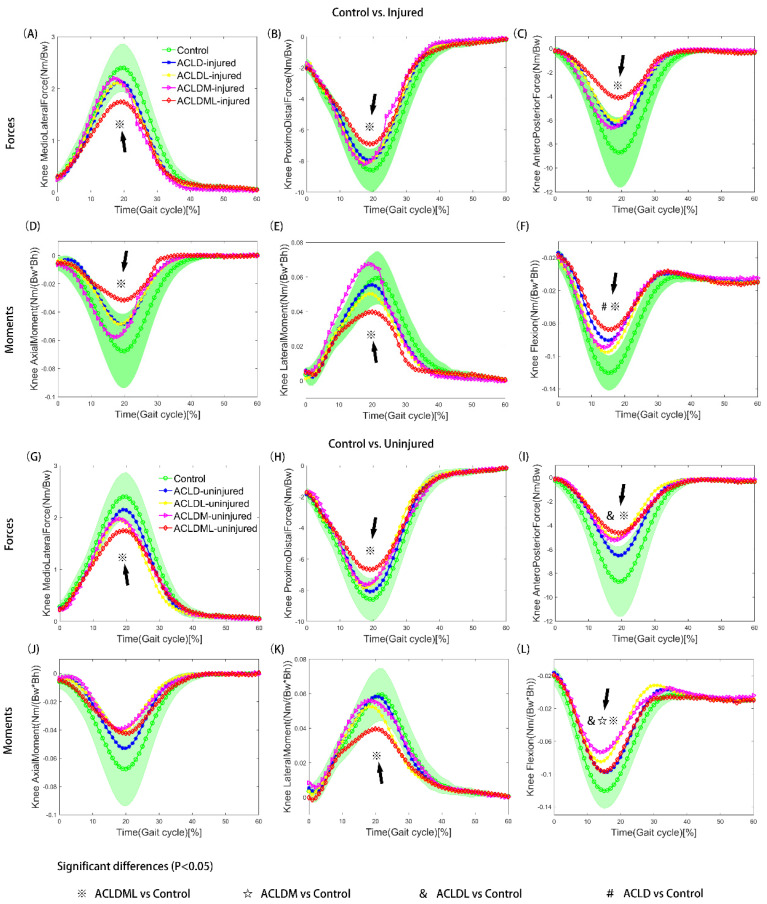
Tibiofemoral joint knee contact forces (**A**−**C**, **G**−**I**) and moments (**D**−**F**, **J**−**L**) of control and patient (ACLD, ACLDL, ACLDM and ACLDML) groups. Comparisons were conducted between control and injured limbs (**A**−**F**) as well as between control and uninjured limbs (**G**−**L**). Segments with significant statistical differences between every two groups were marked with symbols. The green shaded area represents the means ± standard deviation of the control group. ACLD, anterior cruciate ligament deficiency; ACLDL, ACLD combined with lateral meniscal injury; ACLDM, ACLD combined with medial meniscal injury; ACLDML, ACLD combined with lateral and medial meniscal injury. Bw: body weight; Bh: body height.

**Table 1 bioengineering-09-00716-t001:** Participant characteristics in each group *.

Parameters	Control	ACLD	ACLDL	ACLDM	ACLDML
Age (years)	29.33 ± 5.79	27.50 ± 1.98	26.60 ± 3.85	29.60 ± 5.41	27.71 ± 6.75
Height (cm)	171.83 ± 3.69	178.08 ± 8.28	179.80 ± 4.87	178.20 ± 7.09	180.29 ± 7.30
Weight (kg)	73.9 ± 6.75	82.58 ± 11.82	88.78 ± 23.46	85.70 ± 11.52	83.71 ± 14.37
BMI (kg/m^2^)	25.09 ± 2.90	26.01 ± 2.90	27.26 ± 5.82	26.97 ± 3.01	25.64 ± 3.29
Pace(m/s)	2.33 ± 0.16	2.40 ± 0.22	2.47 ± 0.28	2.13 ± 0.28	2.42 ± 0.17
Tegner score	/	3.27 ± 0.75	4.60 ± 3.01	3.60 ± 1.02	5.33 ± 1.20
Time since injury (months)	/	10.50 ± 6.47	9.60 ± 3.29	15.40 ± 8.11	14.14 ± 5.37

* Data are presented as mean ± standard deviation. ACLD, anterior cruciate ligament deficiency; ACLDL, ACLD combined with lateral meniscal injury; ACLDM, ACLD combined with medial meniscal injury; ACLDML, ACLD combined with lateral and medial meniscal injury.

**Table 2 bioengineering-09-00716-t002:** The absolute value of Cohen’s d effect size compared to the control group *.

	ACLD		ACLDL		ACLDM		ACLDML	
	Injured	Uninjured	Injured	Uninjured	Injured	Uninjured	Injured	Uninjured
Maximum ML force	0.18	0.49	0.70	**1.05**	0.50	**0.94**	**1.79**	**1.60**
Minimum PD force	0.42	0.34	0.39	0.61	0.31	0.67	**1.51**	**1.42**
Minimum AP force	0.44	0.59	**0.85**	**1.27**	0.59	**1.20**	**1.90**	**1.54**
Minimum axial moment	0.62	0.58	0.80	**1.05**	0.42	**1.17**	**1.45**	**1.00**
Maximum lateral moment	0.20	0.07	0.71	0.51	0.61	0.30	**1.52**	**1.64**
Minimum Flexion moment	**1.13**	**1.05**	**1.14**	**1.54**	**1.54**	**2.28**	**1.98**	**0.95**

* Small effect size: 0.2≤d<0.5; middle effect size: 0.5≤d<0.8; large effect size: d>0.8 ACLD, anterior cruciate ligament deficiency; ACLDL, ACLD combined with lateral meniscal injury; ACLDM, ACLD combined with medial meniscal injury; ACLDML, ACLD combined with lateral and medial meniscal injury. ML, mediolateral; PD, proximodistal; AP, anteroposterior.

**Table 3 bioengineering-09-00716-t003:** Group Means (Standard Deviations) for kinetic parameters in the control limbs, injured and contralateral uninjured limbs during jogging.

	Control Group	Injured Groups							
		ACLD		ACLDL		ACLDM		ACLDML	
		Injured	Uninjured	Injured	Uninjured	Injured	Uninjured	Injured	Uninjured
**Joint reaction forces/Bw**									
Maximum ML force	2.40(0.43)	2.14(2.16)	2.17(0.52)	2.11(0.35)	1.97(0.33)	2.19(0.38)	1.97(0.54)	1.74(0.15) *	1.74(0.37) *
Minimum PD force	−8.60(1.30)	−7.94(1.87)	−8.10(1.65)	−8.10(1.16)	−7.86(0.88)	−8.21(1.04)	−7.64(1.83)	−6.90(0.54) *	−6.67(1.19) *
Minimum AP force	−7.97(2.36)	−6.48(4.34)	−6.53(2.46)	−6.04(1.90)	−5.09(1.92)*	−6.63(1.97)	−5.23(1.98)	−4.13(0.75) *	−4.65(1.53) *
**Joint moment/(Bw × Bh) (×10^−1^)**									
Minimum axial moment	−0.68(0.26)	−0.49(0.36)	−0.53(0.25)	−0.49(0.13)	−0.41(0.25)	−0.58(0.14)	−0.40(0.14)	−0.32(0.22) *	−0.42(0.26)
Maximum lateral moment	0.60(0.14)	0.56(0.26)	0.59(0.13)	0.50(0.14)	0.53(0.13)	0.68(0.09)	0.56(0.11)	0.40(0.11) *	0.40(0.06) *
Minimum Flexion moment	−1.21(0.21)	−0.82(0.46)*	−0.98(0.23)	−0.95(0.28)	−0.85(0.30)*	−0.89(0.20)	−0.73(0.21) *	−0.68(0.37) *	−0.97(0.33)

* Statistically significant difference (*p* < 0.05) compared with controls from post-hoc pairwise comparison. ACLD, anterior cruciate ligament deficiency; ACLDL, ACLD combined with lateral meniscal injury; ACLDM, ACLD combined with medial meniscal injury; ACLDML, ACLD combined with lateral and medial meniscal injury. ML, mediolateral; PD, proximodistal; AP, anteroposterior; Bw, body weight; Bh, body height.

## Data Availability

Not applicable.

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
