# Peer review of "Identification of Kinetic Abnormalities in Male Patients after Anterior Cruciate Ligament Deficiency Combined with Meniscal Injury: A Musculoskeletal Model Study of Lower Limbs during Jogging"

_bioengineering, 2022, doi:10.3390/bioengineering9110716_

Round 1
Reviewer 1 Report
Dear Authors
I read this paper with interest. I recommend acceptance after some minor edits.
Introduction
I am not sure why you are doing this research. You will show that if there is an injury then the joint forces/moments will be lower, however, this is quiet obvious. How will you apply this information in the clinical setting to help the practitioner?
Methods
Line 108 Use Twente for the name of the model with a capital letter
Results
Excellent
Discussion
Excellent except missing some information about how the results can be applied to help improve rehabilitation training.
Reviewer 2 Report
Thank you for permitting me to review this manuscript
In this study the authors assessed the kinematics and kinetics of ACLD subjects , with meniscal injury (ACLDM° in comparison with ACLD alone and control subject during the stance phase . They found abnormal kinetic alterations at the loading response and mid stance phase in different group of ACLDM patients compared to ACLD alone and control patients.
The results are not surprising
however sample size is very limited and general conclusion cannot be drawn
Please provide reference for by using the motion capture system and subject-specific multi-body dynamic mode since this method should be clinically validated .
the caption systeme need to be more elaborated.
What about the possibilty of injury in non affected limb , please developp a mini paragraph on this issue in the discussion section
Please define more accurately for the readers , kinematics and kinetics since as stated kinematics was not evaluated in this study.
please provide real picture of the system and sensors.
The conclusion is very soft and logical I think it should also focus on methodology
Reviewer 3 Report
Dear authors
The article describes the variation in the kinetic pattern during jogging with and without an ACL- and Meniscal Injury. A topic of interest in predicting the long term outcome of ACL-treatment.
There are some principal questions:
- Subjects:
hours of sport per week,
how did the Injury occur: Through a non-contact or contact accident.
- Technical information about the 3D-motion capture technics ?:
e.g. Sampling rate, Marker size, accuracy of the model (e.g. axis knee joint), error in the kinematic data due to marker placement
- type of force plate, sampling rate,
Questions:
- Table 1: Abbreviations not explained, please move at the end of chapter 2.1
- Text to Figure 1; Line 106/107: how is the inverse dynamic linked between the 2 maker models
- Line 223/ 224: numbers: Unit?
- Line 243: “A literature [26] summarized”: please add name of author.
Round 2
Reviewer 2 Report
The authors have significantly improved the manuscript
Reviewer 3 Report
No further comments